behaviour

defensive contributions, *Neolamprologus pulcher*, outgroup conflict, social interactions, territorial intrusions

**Author for correspondence:**
Ines Braga Goncalves
e-mail: ines.goncalves@bristol.ac.uk

# Experimental evidence that intruder and group member attributes affect outgroup defence and associated within-group interactions in a social fish

Ines Braga Goncalves and Andrew N. Radford

School of Biological Sciences/Life Sciences, University of Bristol, 24 Tyndall Avenue, Bristol BS8 1TQ, UK

IB, 0000-0003-0659-9029; ANR, 0000-0001-5470-3463

In many social species, individuals communally defend resources from conspecific outsiders. Participation in defence and in associated within-group interactions, both during and after contests with outgroup rivals, is expected to vary between group members because the threat presented by different outsiders is not the same to each individual. However, experimental tests examining both the contributions to, and the consequences of, outgroup conflict for all group members are lacking. Using groups of the cichlid *Neolamprologus pulcher*, we simulated territorial intrusions by different-sized female rivals and altered the potential contribution of subordinate females to defence. Dominant females and subordinate females defended significantly more against size- and rank-matched intruders, while males displayed lower and less variable levels of defence. Large and small, but not intermediate-sized, intruders induced increased levels of within-group aggression during intrusions, which was targeted at the subordinate females. Preventing subordinate females from helping in territorial defence led to significant decreases in post-contest within-group and female-specific submissive and affiliative displays. Together, these results show that the defensive contributions of group members vary greatly depending both on their own traits and on intruder identity, and this variation has significant consequences for within-group social dynamics both during and in the aftermath of outgroup contests.

## 1. Introduction

Individuals in many social species live in relatively stable groups, defending important resources from conspecific outsiders [1–3]. Research into outgroup conflict has traditionally focused on actual encounters between rivals, including which individuals contribute to defensive actions, the characteristics of contests and what determines the outcome [4–6]. But there is an increasing awareness that outgroup conflict can impact social interactions between groupmates both during [7,8] and after [9–11] encounters with outsiders or cues of their presence. It is important to consider the influences on within-group behaviour because these are likely to be longer-lasting effects than the period of active defence, and thus are crucial for a full understanding of the costs and benefits of sociality [12]; considering both defensive and within-group behaviour concurrently is important because immediate and subsequent responses to outgroup threats may [13] or may not [9] be tightly coupled. Because the threat presented by different outsiders is not the same, and groups comprise heterogeneous individuals who experience different costs and benefits from outgroup conflict, variation is expected between group members in both contest participation and in the resulting within-group social interactions [6,12]. However, there are relatively few experiments that have tested the causes and magnitude of variation in outgroup defensive contributions [14–17], and most have focused on just a subset of group members.

Moreover, the very few experimental investigations of how out-group contests affect within-group interactions [9,13,18] have all considered the aftermath of contests. To our knowledge, there are no experimental studies that have concurrently assessed the contributions of all group members towards outgroup defence and towards associated within-group social interactions during and after encounters with rivals.

Differences in resident group member attributes and in intruder identity are two factors expected to influence participation in outgroup contests. Groups often contain individuals of different social rank, sex, size, age and relatedness, and these differences can influence motivation to participate in defensive actions [6]. Even when the outgroup threat is to resources of value to all members, such as food or territory, individuals who stand to benefit and/or lose more might be expected to contribute more to defence [1,7,9]. For instance, in vervet monkeys, *Chlorocebus aethiops pygerythrus*, higher-ranked females who have priority access to food resources are more likely than lower-ranked females to participate in contests with neighbouring groups over those resources [7]. Variation in defensive contributions is expected to be even more prevalent in contests against single outsiders seeking reproductive opportunities or group membership, where different intruder attributes, such as sex or size, mean that the threat posed to different group members varies greatly [15,19,20]. For example, male meerkats (*Suricata suricatta*), who may lose dominance or position to prospecting males, invest more in fighting intruders than do females [21].

Individual contributions towards outgroup defence may additionally impact and be impacted by the contributions of groupmates and by within-group interactions during contests. During outgroup contests, individuals may monitor the behaviour of groupmates [19,22] and adjust their own accordingly. For instance, in western bluebirds, *Sialia mexicana*, individual levels of territory defence towards single intruders correlate negatively with group size, indicating that individuals benefit from load-lightening while collectively maintaining similar defence levels as smaller groups [19]. Moreover, where relative participation rather than relative group size probably determines victory against outsiders [23], social incentives (e.g. aggression or affiliation) may be used by more-invested members to recruit new participants and/or reward existing ones, as indicated in recent observational work on vervet monkeys [7]. However, what is lacking are experimental tests of how group members interact with each other during contests with outsiders that present different threats.

There is growing evidence that within-group affiliative interactions can increase following contests with rival groups [9,11–13]. For instance, in green woodhoopoes, *Phoeniculus purpureus*, dominant individuals spend more time allopreening subordinates following prolonged contests, particularly if the group lost the contest [11], and after contests with unfamiliar groups, who pose a greater threat of territory usurpation than do neighbours [13]. In principle, negative social incentives might also be used to manipulate uncooperative group members into increasing their defensive contributions in future conflicts [12]. However, empirical evidence of increased within-group aggression following outgroup conflict is currently lacking.

The daffodil cichlid, *Neolamprologus pulcher*, provides an ideal opportunity to address questions relating to outgroup conflict and potential within-group consequences because it is a territorial, cooperatively breeding species with frequent interactions between group members [24]. Social groups comprise a dominant breeding pair and 0–20 smaller subordinate helpers of both sexes [25]. Defence against conspecific intruders is an important helping behaviour performed by all subordinates, albeit at varying levels [24]. Previous work in which focal observations were done on subsets of group members has shown that intruder sex and size, resident dominant and subordinate sex (relative to the intruder), and subordinate size may influence participation in outgroup contests [14,15]. Furthermore, group members may increase their defensive efforts (i.e. compensate) when faced with defecting groupmates [18]. Subordinates disperse to take over breeding or higher-ranked subordinate positions in other groups [26]. Therefore, territorial intrusions threaten either a breeding or a subordinate position [27], but attempts to usurp a breeding position are probably more disruptive to all members [28]. Within-group aggression, submission and affiliation are used to mediate social relationships [25,29]. Aggressive behaviours function to establish, reinforce and stabilize social hierarchies [28], submissive displays are used to reduce aggression [18,30], and affiliation is exchanged to promote intragroup cooperation [9,12]. One previous experimental study found increases in overall levels of within-group affiliation in the immediate aftermath of intrusions by rival groups [9].

In *N. pulcher*, subordinate females (SFs) contribute more to territorial defence [14] and are more likely to inherit the breeding position than male subordinates [26,28], and more helpful females are more likely to breed as subordinates [31]. However, the presence of large female subordinates is associated with increased conflict between the breeders and with reductions in dominant female (DF) growth [32]. As such, while female subordinates incur important direct fitness benefits from maintaining their group membership, DFs should evict less helpful females to minimize reproductive costs, creating a dynamic where close social monitoring and enhanced responsiveness between females might be expected. In our study, we therefore used unfamiliar female intruders and focused on the effects of outgroup contests on within-group behaviours displayed and received by each group member during and after intrusions, and more specifically on their impact on DF–SF social dynamics.

We used experiments to test how female intruder identity affects individual contributions to territorial defence and to within-group interactions during contests (Experiment I), how SF participation in contests affects the contributions of others to territorial defence (Experiment II), and how both affect post-contest within-group interactions (Experiments I and II). We considered the defensive and within-group behaviours of all group members, but also made some specific predictions. When faced with female intruders of different sizes, we expected resident females to defend most aggressively against size-matched individuals (prediction 1) and resident males to show lower and less variable levels of defensive behaviour than females (prediction 2). A small number of previous *N. pulcher* studies have considered how a subset of group members differ in their defensive responses to a particular intruder [14,15,18], but have not experimentally tested differences between all group members in response to different-sized intruders. When a group member was experimentally prevented from helping against a dominant-sized intruder, we expected other group members to display compensatory defensive efforts (prediction 3), as was previously shown against subordinate-sized intruders [18]. During contests, we expected

variation in the amount of within-group aggression, affiliation and submission depending on intruder size (prediction 4); for instance, we expected intrusions by dominant-sized females to be more disruptive to groups and lead to increases in dominant, particularly female, aggression as a potential participant-recruitment strategy [7]. No previous experiments have tested how within-group interactions are affected during an intrusion period. We also expected changes in within-group interactions following contests [12]; the few previous experiments testing these ideas have focused on the collective threat posed by other groups [9–11,13] rather than that arising from the intrusion of individuals. Specifically, in terms of intruder-size variation, we expected dominant-sized female intruders to have a greater impact than smaller intruders on affiliation between group members in general [9,13], and between the two resident females in particular (prediction 5). Moreover, we expected SFs to use affiliative and submissive behaviours towards DFs as appeasement strategies following intrusions by smaller conspecifics, who threaten their specific position in the group (prediction 6) [33]. With respect to variation in SF contributions to defence, we expected breeders, particularly the female, to act affiliatively towards cooperative subordinates and aggressively towards uncooperative ones (prediction 7) [22] and for subordinates to exhibit appeasement behaviour (increased affiliation and submission) following displays of perceived uncooperative behaviour (prediction 8) [18,22].

## 2. Methods

### (a) Study species and husbandry

Using a captive population of *N. pulcher* housed at the University of Bristol, we formed groups of four individuals, comprising a breeding pair and a helper of each sex, ensuring that each dominant individual was at least 5 mm larger than the same-sex subordinate to aid identification and reduce within-sex aggression. Each group was housed in a 70 l tank that formed its territory (full details in electronic supplementary material, Methods).

### (b) Experimental protocol

We conducted two experiments to investigate the impacts of intrusions by single unfamiliar conspecifics on individual participation in territorial defence and on within-group social interactions. In Experiment I, we considered how different-sized female intruders affect the defence intensity of group members and the within-group social interactions both during and after the intrusion. In Experiment II, we considered how variation in the contribution of SFs to defence affects the contributions of the other group members and post-intrusion behavioural interactions. For each experiment, we conducted three simulated intrusions 24 h apart in a counterbalanced order (full details in electronic supplementary material, Methods). Following a 10 min pre-intrusion observation period, we added an intruder (which had previously been netted out of her home tank and held in a container for 10 min), to a side compartment of the focal tank, obscured from view of the resident group by an opaque partition. After a 5 min settling period, we removed the opaque partition to initiate the 10 min intrusion observation period. At the end of the intrusion, we removed the intruder and transferred her back to her home tank and had a 10 min post-intrusion observation period.

In Experiment I ($n = 12$ groups), we used unfamiliar females of different sizes to vary the threat posed. The Large intruder was size-matched to the DF, the Medium intruder was size-matched to the SF and the Small intruder was smaller than both females. We used 30 females for the 36 simulated intrusions (one Large,

three Medium and two Small females were used as intruders twice each). In Experiment II ($n = 14$ groups), we manipulated SF ability to observe (and thus their knowledge of) the intrusions and to participate in defence (and thus their level of cooperation). We used unfamiliar females, matched in size to the resident DF as intruders; no female was used as an intruder for more than one focal group, but each group received the same female intruder in all three trials. Prior to an intrusion, the SF was isolated behind a transparent partition on the opposite side of the tank to the intrusion compartment. The intruder was then introduced and left to settle for 5 min as in Experiment I. In the Cooperative treatment, the transparent partition separating the SF was removed so she could return to the group just prior to the start of the intrusion; thus, the subordinate could participate in defence. In the Uncooperative treatment, the SF was kept behind the transparent partition so that she could observe the intrusion but could not take part in defence. Although not formally quantified, all SFs spent time watching the intrusions and their groupmates, and all attempted to join their group by swimming along the sides of the barrier and nudging against it. In the Unaware treatment, visual contact between the subordinate and the rest of the group was blocked by adding an opaque partition prior to the start of the intrusion, so that the SF could neither observe the intrusion nor help with defence. At the end of the intrusion, we removed the intruder and allowed the SF (in the Uncooperative and Unaware treatments) to return to the group by removing the relevant partitions. In both experiments, intruders were selected from other (focal and non-focal) groups; individuals were not used as intruders in the same week that their group was used as the resident focal group (further details in electronic supplementary material, Methods). All fish were retained as part of the captive study population at the end of each experiment.

All experimental trials were video recorded (Sony Handycam HDR-XR520) as three separate 10 min observation periods: an undisturbed pre-intrusion period; the intrusion period and the post-intrusion period. Using JWatcher (v. 1.0) and following previously established behavioural protocols for this species [25,34,35], we recorded frequencies of aggression, submission and affiliation displayed and received by each group member and frequencies of the same behaviours exchanged between the females in the group (see electronic supplementary material, Methods). From the intrusion period, we also recorded the frequency of aggressive behaviours performed by all group members towards the intruder and intruder responsiveness towards the resident group.

### (c) Statistical analyses

Data from the three observation periods in both experiments were analysed using linear mixed-effects models (LMMs). In all analyses, our main factors of interest were treatment (Experiment I: Large, Medium or Small intruder; Experiment II: Cooperative, Uncooperative or Unaware SF) and its interaction with individual category (DF, dominant male (DM), SF, subordinate male (SM)). When this interaction was significant, we analysed the effects of treatment on each individual category separately (data subsets). We controlled for various other fixed and random factors in each case (full details in electronic supplementary material, Methods). We report only statistically significant effects of these main factors of interest in the Results, but full information on all models and removed terms are presented in the electronic supplementary material, Results. Analyses of data subsets and post hoc comparisons are reported in the text.

To assess defence behaviour of resident group members, we analysed frequencies of defensive acts against the intruder. To investigate within-group interactions during the intrusions in Experiment I, we assessed the frequency of aggression, submission and affiliation displayed (independent of receiver) and received

(independent of giver) and the frequencies of these behaviours exchanged directly between the DF and the SF. We assessed absolute frequencies of behavioural displays during this observation period, rather than changes in frequencies relative to pre-intrusion levels, because the presence of the intruder during the intrusion period but not during the pre-intrusion observation period means they are not directly comparable. We did not assess within-group interactions during the intrusions in Experiment II because the Uncooperative treatment precluded the females from interacting to the same extent as in the Cooperative treatment, and the Unaware treatment prevented them from interacting entirely with their groupmates. To investigate post-intrusion within-group interactions in both experiments, we analysed changes in frequencies of aggression, submission and affiliation displayed and received from the pre-intrusion to the post-intrusion observation period (no intruder was present in both periods) by all group members. We also analysed changes in DF aggression and affiliation directed at the SF, and changes in SF affiliation and submission directed at the DF.

## 3. Results

### (a) Territorial defence behaviour

#### (i) Experiment I (predictions 1 and 2)

The amount of aggression directed at intruders was significantly affected by the interaction between intruder size (treatment) and individual category (LMM: $\chi^2 = 37.60$, d.f. = 6, $p < 0.001$, electronic supplementary material, table S1; figure 1). Treatment significantly affected the defence behaviour of both types of female (DF: $\chi^2 = 14.67$, d.f. = 2, $p < 0.001$; SF: $\chi^2 = 17.28$, d.f. = 2, $p < 0.001$), but not that of the males (DM: $\chi^2 = 1.40$, d.f. = 2, $p = 0.496$; SM: $\chi^2 = 2.26$, d.f. = 2, $p = 0.323$). Specifically, DFs were more aggressive towards size-matched (Large) intruders than towards Medium (post hoc paired $t$-test: $t = 2.22$, d.f. = 11, $p = 0.048$) and Small ($t = 3.51$, d.f. = 11, $p = 0.005$) intruders, and showed similar levels of aggression towards Medium and Small intruders ($t = 1.62$, d.f. = 11, $p = 0.134$). SFs were more aggressive towards size-matched intruders (Medium) than to the other intruder categories (Large: $t = 3.13$, d.f. = 11, $p = 0.010$; Small: $t = 2.67$, d.f. = 11, $p = 0.022$), but also more aggressive towards Small intruders than Large ones ($t = 3.24$, d.f. = 11, $p = 0.008$).

#### (ii) Experiment II (prediction 3)

Defence behaviour towards intruders was not significantly affected by either treatment or its interaction with individual category, after controlling for significant effects of group size and individual category (electronic supplementary material, table S2).

### (b) Within-group interactions during intrusions

#### (i) Experiment I (prediction 4)

Within-group displays of aggression were significantly affected by treatment (LMM: $\chi^2 = 7.94$, d.f. = 2, $p = 0.019$, figure 2a) after controlling for a significant effect of individual category (electronic supplementary material, table S3a). Overall, higher levels of aggression were displayed during intrusions by Large females compared with Medium females (post hoc paired $t$-test: $t = 2.53$, d.f. = 47, $p = 0.015$), but not Small females ($t = 0.95$, d.f. = 47, $p = 0.349$); there was also more within-group aggression displayed during intrusions by Small intruders than Medium ones ($t = 2.78$, d.f. = 47, $p =$

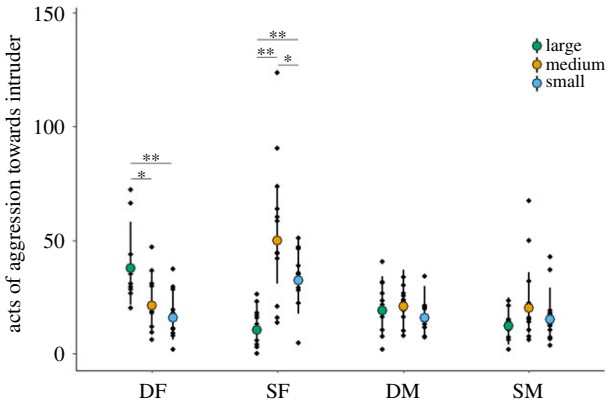

**Figure 1.** Defence behaviour of resident group members towards different-sized female intruders (Experiment I). Fitted values (mean ± 95% confidence intervals) and partial residuals (black dots) from LMMs in electronic supplementary material, table S1 are shown. Statistics run on square-root transformed data. Significant treatment differences within individual categories highlighted. *$p < 0.05$, **$p < 0.01$. DF, dominant female; DM, dominant male; SF, subordinate female; SM, subordinate male. (Online version in colour.)

0.008). Levels of aggression received during intrusions were dependent on the interaction between treatment and individual category (LMM: $\chi^2 = 18.27$, d.f. = 6, $p = 0.006$, electronic supplementary material, table S3b, figure 2b). Specifically, only the frequency of aggression received by SFs was significantly affected by treatment ($\chi^2 = 6.71$, d.f. = 2, $p = 0.035$), with SFs receiving more aggression during intrusions by Large females than during intrusions by Medium females (post hoc paired $t$-test: $t = 2.25$, d.f. = 11, $p = 0.046$), but not those by Small females ($t = 1.57$, d.f. = 11, $p = 0.146$); SFs received similar amounts of aggression during intrusions by the two smaller intruder types ($t = 1.97$, d.f. = 11, $p = 0.074$).

Neither within-group submission displayed nor received were significantly affected by either treatment or its interaction with individual category, after controlling for significant effects of order and individual category (electronic supplementary material, table S4). Neither within-group affiliation displayed nor received were significantly affected by either treatment or its interaction with individual category, after controlling for a significant effect of individual category (electronic supplementary material, table S5).

DFs did not significantly adjust the amount of aggression they directed at SFs during intrusions in response to intruder identity (electronic supplementary material, table S6a). Similarly, the amount of submissions SFs directed at DFs during intrusions was not significantly affected by treatment (electronic supplementary material, table S6b). By contrast, the amount of affiliation DFs directed at SFs depended significantly on the interaction between treatment and intruder responsiveness (LMM: $\chi^2 = 10.33$, d.f. = 2, $p = 0.006$; electronic supplementary material, table S6c). DFs reduced the amount of affiliation directed towards SFs significantly with increasing levels of intruder responsiveness only during intrusions by Large females (ANCOVA: $F_{1,10} = 6.00$, $p = 0.034$). There was no significant effect of treatment on the amount of affiliation directed by SFs towards DFs, after controlling for a significant effect of intruder responsiveness (electronic supplementary material, table S6d).

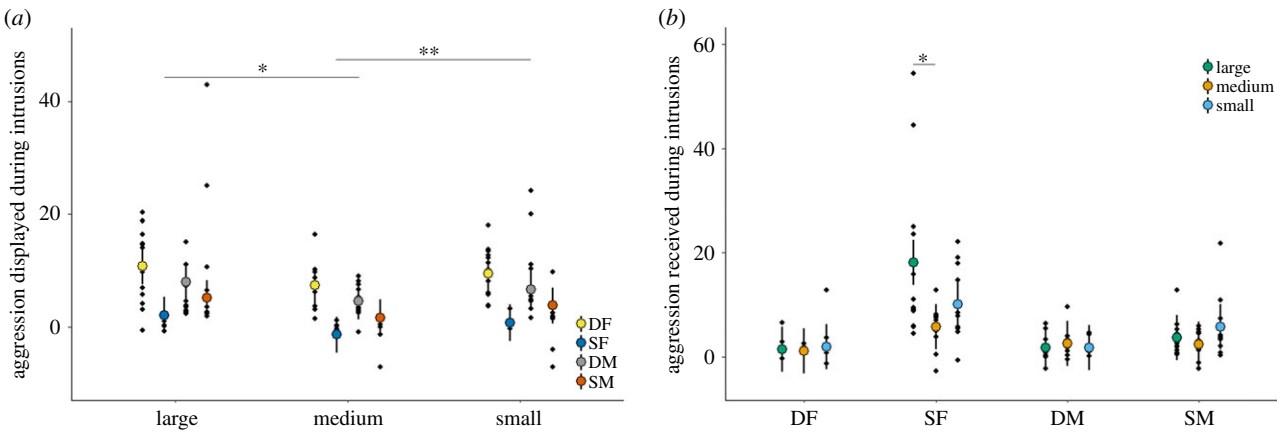

**Figure 2.** Within-group acts of aggression (*a*) displayed and (*b*) received during intrusions by different-sized females (Experiment I). Fitted values (mean ± 95% confidence intervals) and partial residuals (black dots) from LMMs in electronic supplementary material, table S3 are shown. Significant treatment differences highlighted. *$p < 0.05$, **$p < 0.01$. DF, dominant female; DM, dominant male; SF, subordinate female; SM, subordinate male. (Online version in colour.)

## (c) Post-intrusion within-group changes in behaviour

### (i) Experiment I (predictions 5 and 6)

Changes in within-group aggression, submission and affiliation displayed and received were not significantly affected by intruder identity nor by its interaction with individual category (electronic supplementary material, tables S7–S9).

DFs did not significantly change their aggressive (electronic supplementary material, table S10a) or affiliative (electronic supplementary material, table S10b) behaviours towards SFs differently following the different intrusions. SFs did not significantly adjust their submissive behaviour towards DFs in response to intruder identity (electronic supplementary material, table S10c). However, treatment had a significant effect on SF affiliative behaviour towards the DF ($\chi^2 = 8.35$, d.f. = 2, $p = 0.015$, electronic supplementary material, table S10d), due to a tendency for SF affiliation directed at DFs to increase following intrusions by Small intruders relative to Large (post hoc paired *t*-test: $t = 2.50$, d.f. = 11, $p = 0.065$) and Medium ($t = 2.021$, d.f. = 11, $p = 0.068$) intruders. This slight increase in affiliation in the Small intruder treatment was not significantly different from zero (one-sample *t*-test: $t = 2.04$, d.f. = 11, $p = 0.066$). Large and Medium intruders did not elicit significantly different changes in SF affiliation directed at DFs ($t = 0.92$, d.f. = 11, $p = 0.380$).

### (ii) Experiment II (predictions 7 and 8)

Neither the level of within-group aggression displayed nor received was significantly affected by either treatment or its interaction with individual category, after controlling for a significant effect of individual category in the latter case (electronic supplementary material, table S11).

Within-group submissive displays were significantly affected by treatment (LMM: $\chi^2 = 7.29$, d.f. = 2, $p = 0.026$; figure 3a), after controlling for a significant effect of individual category (electronic supplementary material, table S12a). Group members increased submissive behaviour following the Cooperative treatment, where all members could participate in defence, compared with the Uncooperative treatment, where the SF could watch but not participate in defence (post hoc paired *t*-test: $t = 2.73$, d.f. = 48, $p = 0.009$). Change in submissive displays was not significantly different between the Unaware treatment and either the Cooperative ($t = 1.60$, d.f. = 48, $p = 0.115$) or the Uncooperative ($t = 1.10$, d.f. = 48, $p = 0.279$)

treatments. Submissions received changed significantly in response to an interaction between treatment and individual category (LMM: $\chi^2 = 13.41$, d.f. = 6, $p = 0.037$; figure 3b), after controlling for a significant effect of trial order (electronic supplementary material, table S12b). This interaction was driven by DFs ($\chi^2 = 6.46$, d.f. = 2, $p = 0.040$), who received significantly more submissions following the Cooperative treatment relative to the Uncooperative (post hoc paired *t*-test: $t = 3.10$, d.f. = 12, $p = 0.009$) but not the Unaware ($t = 1.64$, d.f. = 12, $p = 0.128$) treatment, with the Uncooperative and Unaware treatments resulting in similar levels of submission received ($t = 0.55$, d.f. = 12, $p = 0.592$). Submissions received by the remaining individual categories were not significantly affected by treatment (LMM, DM: $\chi^2 = 2.19$, d.f. = 2, $p = 0.335$; SF: $\chi^2 = 4.08$, d.f. = 2, $p = 0.130$; SM: $\chi^2 = 1.27$, d.f. = 2, $p = 0.530$).

Within-group affiliation displays changed significantly in response to treatment (LMM: $\chi^2 = 6.46$, d.f. = 2, $p = 0.040$; figure 3c), after controlling for a significant effect of individual category (electronic supplementary material, table S13a). Intrusions in the Cooperative treatment resulted in higher levels of within-group affiliation displays than following the Unaware treatment (post hoc paired *t*-test: $t = 2.25$, d.f. = 48, $p = 0.029$), with the Uncooperative treatment presenting intermediate affiliation levels (Cooperative cf. Uncooperative: $t = 1.28$, d.f. = 48, $p = 0.206$; Uncooperative–Unaware: $t = 1.69$, d.f. = 48, $p = 0.097$). Changes in affiliation received were significantly affected by the interaction between treatment and individual category (LMM: $\chi^2 = 14.16$, d.f. = 6, $p = 0.025$; figure 3d), after controlling for a significant effect of trial order (electronic supplementary material, table S13b). SFs were significantly affected by treatment ($\chi^2 = 16.07$, d.f. = 2, $p < 0.001$): they received less affiliation following the Unaware treatment relative to the other treatments (post hoc paired *t*-tests, Cooperative cf. Unaware: $t = 3.84$, d.f. = 12, $p = 0.002$; Uncooperative cf. Unaware: $t = 2.76$, d.f. = 12, $p = 0.017$), and following the Uncooperative relative to the Cooperative treatment ($t = 2.50$, d.f. = 12, $p = 0.028$). None of the other individual categories were affected significantly by treatment (LMMs, DF: $\chi^2 = 4.07$, d.f. = 2, $p = 0.131$; DM: $\chi^2 = 0.04$, d.f. = 2, $p = 0.979$; SM: $\chi^2 = 3.11$, d.f. = 2, $p = 0.211$).

DFs did not significantly change their aggressive behaviour towards SFs in response to treatment (electronic supplementary material, table S14a), but they did change their affiliative behaviour ($\chi^2 = 7.33$, d.f. = 2, $p = 0.026$, electronic

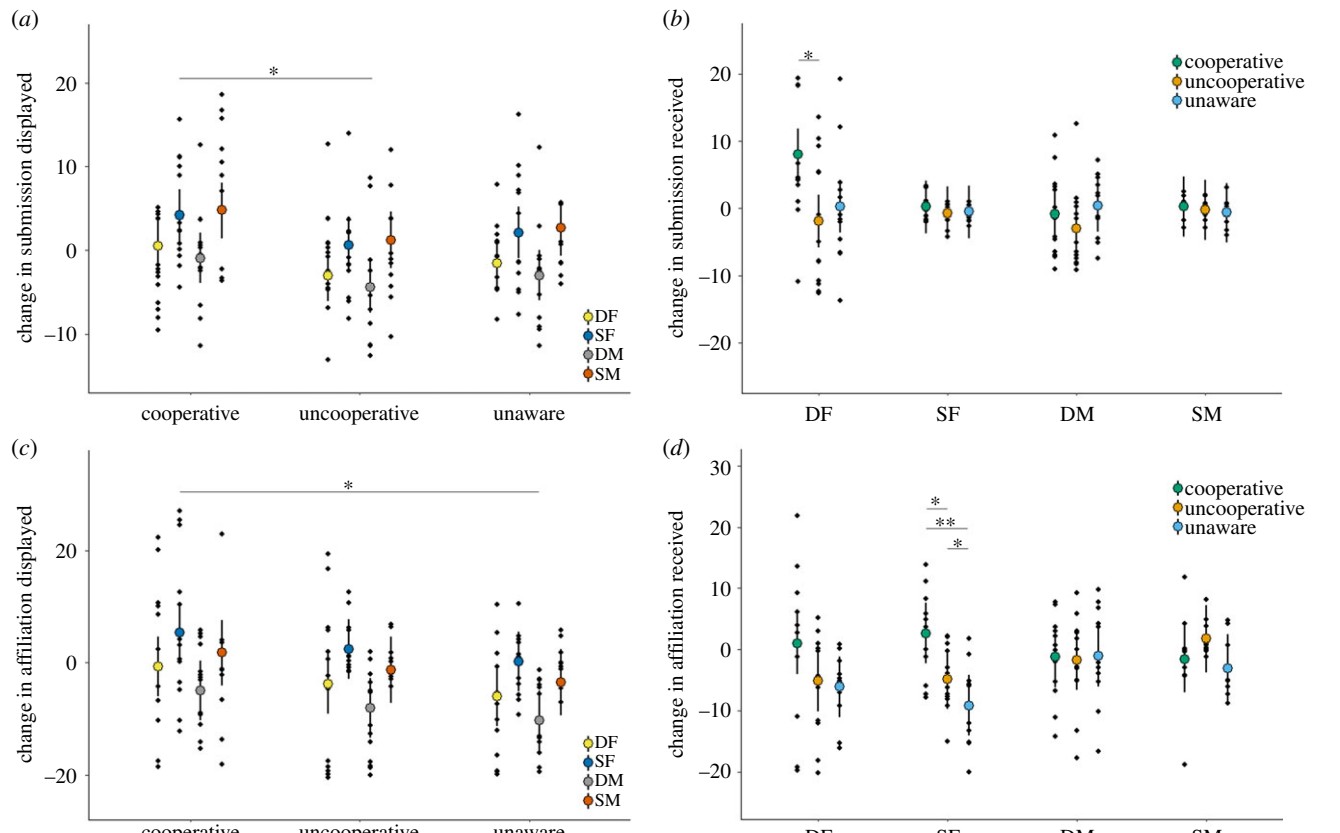

**Figure 3.** Post-intrusion change in within-group frequency of (*a*) submission displayed, (*b*) submission received, (*c*) affiliation displayed and (*d*) affiliation received depending on variation in subordinate female presence or absence and contribution to territorial defence (Experiment II). Fitted values (mean ± 95% confidence intervals) and partial residuals (black dots) from LMMs in electronic supplementary material, tables S12 and S13 are shown. Significant treatment differences highlighted. \*$p < 0.05$, \*\*$p < 0.01$. DF, dominant female; DM, dominant male; SF, subordinate female; SM, subordinate male. (Online version in colour.)

supplementary material, table S14b). DFs reduced their levels of affiliation towards SFs significantly following the Unaware treatment compared with the Cooperative (post hoc paired *t*-test: $t = 2.45$, d.f. = 12, $p = 0.030$) and the Uncooperative ($t = 3.02$, d.f. = 12, $p = 0.011$) treatments; a change that led to an absolute reduction in DF affiliative behaviours after the Unaware treatment (one-sample *t*-test: $t = 2.72$, d.f. = 12, $p = 0.019$; electronic supplementary material, figure S1a).

SFs did not significantly adjust their submissive behaviour towards DFs differently in response to the different treatments, after controlling for a significant effect of trial order (electronic supplementary material, table S14c). By contrast, SFs did significantly adjust their affiliative behaviour towards DFs in response to treatment ($\chi^2 = 6.33$, d.f. = 2, $p = 0.042$; electronic supplementary material, table S14d and figure S1b), with SFs directing less affiliation at DFs following the Unaware treatment relative to the Uncooperative treatment (post hoc paired *t*-test: $t = 3.23$, d.f. = 12, $p = 0.007$) and to the Cooperative treatment, though not significantly so ($t = 2.17$, d.f. = 12, $p = 0.051$). The change in affiliation in the Unaware treatment was significantly different from zero (one-sample *t*-test: $t = -2.83$, d.f. = 12, $p = 0.015$) indicating an absolute reduction in SF affiliation directed at DFs. Although changes in SF affiliation directed at DFs were not significantly different between the Cooperative and Uncooperative treatments (post hoc paired *t*-test: $t = 0.21$, d.f. = 12, $p = 0.841$), affiliative displays increased significantly following the Uncooperative intrusions relative to pre-intrusion levels (one-sample *t*-test: $t = 2.82$, d.f. = 12, $p = 0.016$), but not following intrusions in the Cooperative treatment ($t = 1.03$, d.f. = 12, $p = 0.321$).

## 4. Discussion

Territorial intrusions by single unfamiliar conspecifics induced defensive behaviour in all resident group members, but we found consistent differences in individual participation that were influenced both by intruder identity and the attributes of the group members. When we manipulated intruder size but not the defensive contributions of resident group members (Experiment I), we found variation in the levels of within-group aggression and affiliation during the intrusions but not in the aftermath of the outgroup contest. By contrast, when we manipulated SF contributions to defence and her ability to interact with group members during the intrusions (Experiment II), we observed significant changes in post-contest affiliative and submissive social interactions, both at the group level and specifically between the dominant and SFs. Our work therefore provides the first experimental evidence of factors driving not only differences in the contributions of all group members to defence against conspecific outsiders, but also variation in within-group social interactions both during and in the immediate aftermath of outgroup contests.

### (a) Territorial defence behaviour

The amount of defensive behaviour directed at intruders depended both on intruder size and on resident group member sex and rank. These results are in line with findings from this [14,15] and other [1,17,19,21] species, where greater defensive effort by group members of the same sex and rank as the intruders has been reported. In Experiment I, female but not male subordinates were highly aggressive towards

size-matched and smaller female intruders, and DFs were consistently the most responsive individuals towards large female intruders (as per predictions 1 and 2). Contrary to prediction 3, we did not find evidence of defensive compensation: other group members did not increase their defensive efforts when defence by the SFs was experimentally reduced in Experiment II. This result contrasts Bergmuller & Taborsky [18], where *N. pulcher* group members increased their defence when a subordinate was prevented from helping. However, the intruder in that study was size-matched to the subordinate, whereas the intruder in our experiment was size-matched to the DF. Therefore, the subordinate prevented from helping by Bergmuller & Taborsky [18] was expected to be a top contributor to defence, while the DF was expected to make the greatest contribution in our experiment. Our results suggest that single *N. pulcher* intruders are generally not perceived as a serious threat to the whole group, even though there may be direct (due to injury or death) or indirect (e.g. due to changes in within-group interactions) effects of the intruder's presence, as well as potential consequences if a takeover or immigration attempt is successful [28]. Instead, defensive decisions appear to be mainly based on individual assessments of the threat level posed by the intruder to each group member's social position [14,15].

## (b) Within-group interactions during intrusions

We found considerable differences in the behavioural interactions between resident group members during the intrusions in Experiment I. Variation in affiliation and in submission displayed and received across the groups reflected differences in social rank, with subordinates generally on the giving and dominants on the receiving end of the continuum, but were not affected by our treatments (contrary to prediction 4). However, in line with prediction 4, within-group aggression displayed and received during intrusions was affected by intruder identity. Overall, group members were more aggressive during intrusions by Large and by Small intruders, compared with those by Medium-sized ones; these effects were mirrored in the aggression levels received by SFs. The effect on dominant aggression of two very different-sized intruders suggests that it may serve different functions depending on context. Within-group aggression during intergroup conflicts acts as a social incentive to increase defensive contributions in vervet monkeys [7]. Likewise, when intruded by large competitors, dominant aggression towards female subordinates, who tend to provide more help than males in *N. pulcher* [15,30], may function to increase immediate defensive effort [36]; explicitly testing this hypothesis about relative changes in defensive contributions in response to within-group behaviour would require further experiments. Alternatively, heightened within-group dominant aggression may result from overall higher aggressive motivation when faced with size-matched intruders. Elevated aggression when social groups face small intruders may also reflect genuine antagonism towards subordinates as *N. pulcher* dominants become less tolerant of own subordinates when potential new subordinates are available [33,37]. This situation is not dissimilar to our Small intruder treatment, where a small female outsider may be more appealing as a helper than a larger female who can become a reproductive competitor [38].

Within-group interactions during outgroup contests are likely to play an important role in successful conflict resolution and consequently influence group dynamics and stability.

Observational work on vervet monkeys has shown that social interactions between group members during intergroup contests can impact group coordination and successful co-defence of resources [7], and help prevent conflict escalation [8]. Future experimental research could profitably explore how conflict-induced changes in within-group interactions influence subsequent defensive contributions of those who, for instance, receive affiliation or aggression, and how long those behavioural incentives may last.

## (c) Post-intrusion within-group changes in behaviour

Overall, there was no change in the frequency or type of post-contest within-group interactions relative to pre-intrusion levels in response to the different-sized intruders (Experiment I); there was no support for predictions 5 and 6. This finding differs markedly from the significant increases in post-intrusion within-group affiliation found in the same study species by Bruintjes *et al.* [9]. Two main methodological differences between the studies might explain this difference in results. First, in Bruintjes *et al.* [9], three simultaneous intruders (probably each posing a direct threat to a different resident group member) were presented, while we simulated single-rival intrusions (probably posing a direct threat to the position of only one group member). Moreover, when presented with multiple intruders, residents potentially had less time to interact with each other than they did in our Experiment I. Within-group interactions during intrusions were not recorded in Bruintjes *et al.* [9], but if such interactions are important for the maintenance of group stability and social dynamics and these were reduced, then there may have been an increased need for social interactions in the aftermath of rival-group intrusions (see discussion on Experiment II below). The second methodological difference is that we simulated intrusions at the edge of the resident group's territory, whereas Bruintjes *et al.* [9] presented intruders at the territory centre just a few centimetres from the breeding shelters. Consequently, in Bruintjes *et al.* [9], the level of threat (core territory position, three intruders) was probably higher than in our manipulation and might have driven greater changes in within-group behaviour [4,13].

Within-group affiliation changed in response to experimental reductions in SF contributions to outgroup defence (Experiment II). When SF was not aware of the intruder and could not participate in defence (Unaware treatment), there were significant reductions in overall and in female-specific affiliative behaviour relative to the Cooperative treatment, where all members could contribute to defence. As affiliative displays function to stabilize social hierarchies, promote reconciliation and are traded to reward cooperation [7,9,39], our results provide several insights into the mediation of social dynamics in this species. First, regarding its function as a reward for defensive effort [9,11,13], we observed the highest levels of affiliation displayed in our Cooperative treatment both at the group level and specifically directed by the DF towards the SF, as per prediction 7. The reverse (i.e. the withdrawal of affiliation) occurred following 'defection' behaviour, with SFs receiving less affiliation when they had not assisted in defence. Second, as an effort to promote reconciliation (prediction 8), the reduction in SF affiliation towards the DF in the Unaware treatment of Experiment II contrasts with the increased affiliation displayed following the Small intruder treatment in Experiment I. However, subordinate affiliation has been shown to correlate positively with participation in group

defence [40] and thus, fewer affiliative displays may be expected in the Unaware treatment as SF could not see nor participate in defence.

Contrary to prediction 7, reductions in SF contributions to defence (Experiment II) did not induce dominant aggression (i.e. dominants did not use aggression as punishment for uncooperative behaviour), at least in the immediate aftermath of conspecific intrusions. The only study to report conclusive evidence of punishment in *N. pulcher* experimentally prevented subordinate help for an extended period (24 h) in groups of different sizes and found resulting increases in dominant aggression only in small groups [22]. Shorter manipulations (10 min to 6 h) conducted in the field [28] or in the laboratory (this study; [18]) did not induce dominant punishment, independent of group size (4–15 adults) and of whether the subordinate was removed from the group (as in our Unaware treatment) or kept in the group but prevented from helping (as in our Uncooperative treatment). Dominant punishment of subordinates therefore seems to be at least partially dependent on group size and potential cognitive constraints at play in larger groups, on the duration of the reduced cooperation and possibly on the opportunities to help (i.e. the availability of cooperative tasks to perform). Regardless, our post-contest result contrasts the observed increases in aggressive interactions during the intrusions, where dominants could have been attempting to stimulate immediate changes in the behaviour of groupmates. Possibly, aggression is employed more commonly to settle immediate disputes, but not extended ones. The time lag between the use of aggression on uncooperative individuals and a future situation when those members are expected to cooperate again may render aggression too costly to be administered effectively [36]. Punishment is not only energetically costly; it can also lead to reduced group cohesion and coordination [40]. Therefore, levels of aggression exhibited may not fully reflect the degree of conflict between individuals [40], and behaviours other than aggression may be used more effectively to punish uncooperative individuals.

Our manipulations induced significant post-intrusion changes in submissive interactions. Contrary to prediction 8, and counterintuitively, given the potential appeasement function of subordinate submission [22], the Uncooperative treatment induced a significant decrease (rather than increase) in SF submissive behaviour relative to the Cooperative treatment. However, the use [18] and usefulness [41] of submission as an appeasement strategy towards dominant individuals have been questioned, with alternative behaviours potentially being used more effectively [18,30]. Although not measured in this study, it is possible that, in the immediate aftermath of a perceived defection, subordinate avoidance of the dominant individuals is a more effective appeasement strategy than submission, particularly in the absence of helping opportunities.

# 5. Conclusion

Outgroup conflict is a common and probably powerful selective force in the animal kingdom, yet experimental tests of its consequences are rare. Our manipulations provide evidence that intrusions by rivals can affect within-group social interactions both during and in the aftermath of outgroup contests. Moreover, we demonstrate that the nature and extent of those interactions can be influenced by the identity of the intruder, by the characteristics (sex and dominance status) of group members and by the contribution of individuals to defensive actions. Considering behaviour directed towards groupmates as well as that towards rivals, by all group members (as we have done here), is necessary for a full understanding of outgroup conflict. Future studies should adopt this approach where feasible, as well as examine longer-term consequences, to help unravel the importance of this relatively neglected aspect of sociality.

**Ethics.** This work was approved by the University of Bristol Ethical Committee (University Investigator Number: UB/16/049).

**Data accessibility.** Data are available from the Dryad Digital Repository: https://doi.org/10.5061/dryad.j5379g6 [42].

**Authors' contributions.** Both authors designed the experiments. I.B.G. performed the experiments and analyses. I.B.G drafted the manuscript and both authors contributed to subsequent writing.

**Competing interests.** We declare we have no competing interests.

**Funding.** This work was supported by a European Research Council Consolidator Grant (project no. 682253) awarded to A.N.R.

**Acknowledgements.** We thank Simon Sanghera for help with the data extraction from videos from Experiment II, Laura Richardson for assistance with the manuscript's figures, and Ben Ashton and two anonymous referees for helpful comments on the manuscript. We thank Michael Taborsky for useful discussions about this project.

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
