## [Reviewer comments · Proceedings of the Royal Society B: Biological Sciences]

Review History

RSPB-2019-1261.R0 (Original submission)

Review form: Reviewer 1

Recommendation

Accept with minor revision (please list in comments)

Scientific importance: Is the manuscript an original and important contribution to its field?

Good

General interest: Is the paper of sufficient general interest?

Good

Quality of the paper: Is the overall quality of the paper suitable?

Good

Is the length of the paper justified?

Yes

Should the paper be seen by a specialist statistical reviewer?

No

Do you have any concerns about statistical analyses in this paper? If so, please specify them explicitly in your report.

No

It is a condition of publication that authors make their supporting data, code and materials available - either as supplementary material or hosted in an external repository. Please rate, if applicable, the supporting data on the following criteria.

Is it accessible?

N/A

Is it clear?

N/A

Is it adequate?

N/A

Do you have any ethical concerns with this paper?

No

Comments to the Author

Goncalves & Radford investigated the within-group consequences of territorial intrusions, and territory defense, against unfamiliar conspecific intruders in the cooperatively breeding cichlid fish *Neolamprologus pulcher*. This was a well-designed study that took an original next step in the study of the relationship between sociality and territory defense. Other studies in this and other systems have investigated the differential response by group members to intruders depending on relative size, status, and sex, but this study is amongst the first to investigate the impact of this defense and variation in it on within-group interactions. While I had several questions regarding the methods and manuscript (see below), I enjoyed reading this well written study, and look forward to seeing future work building upon these findings.

1. The predictions outlined in lines 103-110 make sense in light of “pay to stay” and “pay to reproduce” hypotheses for cooperative breeding in this system (and are supported by the studies the authors reference in this section). Would these predictions hold true under other hypotheses that have been proposed in this system such as “group augmentation” or “signals of prestige”?
2. While its clearly outlined in the supplemental materials, I’d encourage the authors to explicitly state in the methods section that size-matching of fish was based on SL rather than mass.
3. Even though it is somewhat obvious, it might be worthwhile to describe the “intrusion period” and “post intrusion period” as “... observation period” to make it clear this is when the behavioral observations took place.
4. Did these intrusions take place in the groups home tank, or were fish moved for the trial?
5. The authors classified “bumping” as an affiliative behavior. While to my knowledge this isn’t resolved, some other researchers have considered bumping a submissive or conciliatory behavior. How frequent were bumps relative to other behaviors classified as affiliative? Given that the

design of Experiment 2 involves preventing a subordinate female from participating in care, would these other perspectives regarding the context of bumping impact your conclusions?

6. The dominant male and female were very close in size in these groups. In wild groups, males are generally noticeably larger than females. Additionally, Ligoocki et al, 2019, (JEZ-A) found that female size relative to their mates influenced female participation in defense. While size differences appear to have remained consistent across groups, could the fact that males were small relative to dominant females have impacted their involvement in defense? Would dominant or subordinate females be more impacted by the size of males in terms of behavioral compensation?

7. It appears that subordinate females varied much more in their participation in territory defense than other individuals. Was there anything that stood out about individuals who were especially territorial against intruders?

8. What were subordinate females doing during the uncooperative treatment? Were they at the barrier "trying" to defend? Was there variation in this behavior that may relate to post-intruder interactions?

9. On line 403 it appears there is an extra space before "Bruintjes..."

10. Were groups kept together following the observation period? Do you have any insights into the long term consequences of these interactions for group members?

Review form: Reviewer 2

Recommendation

Major revision is needed (please make suggestions in comments)

Scientific importance: Is the manuscript an original and important contribution to its field?

Acceptable

General interest: Is the paper of sufficient general interest?

Acceptable

Quality of the paper: Is the overall quality of the paper suitable?

Acceptable

Is the length of the paper justified?

Yes

Should the paper be seen by a specialist statistical reviewer?

No

Do you have any concerns about statistical analyses in this paper? If so, please specify them explicitly in your report.

Yes

It is a condition of publication that authors make their supporting data, code and materials available - either as supplementary material or hosted in an external repository. Please rate, if applicable, the supporting data on the following criteria.

Is it accessible?

No

Is it clear?

N/A

Is it adequate?

N/A

Do you have any ethical concerns with this paper?

No

Comments to the Author

The study „Experimental evidence that intruder and group member attributes affect out-group defense and associated within-group interactions in a social fish“ describes two very interesting experiments creating different out-group threats (experiment 1) and different options for subordinate group members to participate (or not) in territory defense (experiment 2). In general, I think the two experiments are very interesting and enhance our understanding about individual contributions during out-group encounters. Nevertheless I feel that the presentation could be improved particularly to point out the new insights gained from these studies.

Comments

Introduction

In my point of view the introduction does not make it clear how this study differs from previously conducted studies mainly in the same species. For example, in LL47-49 the authors state that “there are relatively few experiments that have tested the causes and magnitude of variation in out-group defensive contributions [...]”. The cited studies are mainly experiments using *N. pulcher* and I think it would make sense to clearly specify what has been done in this species and what aspects are needed to fully understand individual contribution during defensive interactions. This would make it easier to understand the predictions outlined in LL114-126. It’s not clear which predictions are novel and which ones are based on previous experiments. Furthermore some of the predictions (e.g. 2, and 3) are not directly connected to an out-group context because compensation of other group members should only occur if the lost help is indeed beneficial (as also discussed in the disunion) and aggression towards uncooperative subordinates requires that helping is somehow enforced (e.g. pay-to-stay). Both are existent in this system but it’s not entirely clear to me how this is linked with an out-group threat.

After reading the manuscript it is not entirely clear to me why the two experiments have been presented in combination. I think that the authors should make it clearer what are the expectations and predictions of each experiment and how each experiment advances our knowledge of out-group threats.

Results

The results section was extremely difficult to read and I was honestly struggling to link it with the predictions or understand why each test and analysis has been done. I would suggest to structure the results section according to the predictions and analyze each prediction separately instead of combining the results into the experimental time line (pre-intrusion, intrusion, post-intrusion).

In general it is not clear to me why the authors analyzed received and directed behaviors separately for the whole group. In my point of view this would require to correct for multiple

testing. Also, I feel that a good justification should be given to analyse the results on a group level and then again more specifically on an individual level (i.e. interactions between DF and SF and vice versa).

LL207-209: Unclear how this was analysed.

LL233-235: Submission is normally a response to received aggression and thus it would be interesting to include received aggression as a covariate in the models.

LL209-2012: In general, did the authors use any correction for multiple testing when performing the posthoc- t tests?

LL310-314: I was surprised that DF affiliation was reduced in the unaware treatment. In Hamilton et al 2005 affiliation e.g. soft touches have been interpreted in an aggressive context. Are affiliative behaviours directed from DF to SF and affiliative behaviours directed from SF to DF the same?

Discussion:

Again I think a clear emphasis on what's new and what's different to other experiment is missing throughout the discussion.

LL379-384: In my opinion if DF aggression is used to increase immediate defensive efforts in SF then SF would have also attacked the larger intruder more than any other intruder. However each group member attacked the intruder which posed the highest risk to themselves supporting that higher DF aggression towards SF might be a result of a higher aggressive motivation in DF.

Decision letter (RSPB-2019-1261.R0)

16-Jul-2019

Dear Dr Braga Goncalves:

Your manuscript has now been peer reviewed and the reviews have been assessed by an Associate Editor. The reviewers' comments (not including confidential comments to the Editor) and the comments from the Associate Editor are included at the end of this email for your reference. As you will see, the reviewers and the Editors have raised some concerns with your manuscript and we would like to invite you to revise your manuscript to address them.

Research ethics:

Use of animals and field studies:

Online supplementary material will also carry the title and description provided during submission, so please ensure these are accurate and informative. Note that the Royal Society will not edit or typeset supplementary material and it will be hosted as provided. Please ensure that

the supplementary material includes the paper details (authors, title, journal name, article DOI). Your article DOI will be 10.1098/rspb.[paper ID in form xxxx.xxxx e.g. 10.1098/rspb.2016.0049].

Please submit a copy of your revised paper within three weeks. If we do not hear from you within this time your manuscript will be rejected. If you are unable to meet this deadline please let us know as soon as possible, as we may be able to grant a short extension.

Best wishes,
Dr Daniel Costa
mailto:proceedingsb@royalsociety.org

Associate Editor
Board Member: 1
Comments to Author:

Both reviewers agree that the paper was well-written and well-designed – overall, an interesting contribution. Both reviewers also provided very useful suggestions regarding areas that need clarification and strengthening. I believe that in thoroughly addressing these concerns, the authors could substantially increase the impact of the work. In addition to considering the reviewers' excellent points, I would also like to hear the authors' responses to several other questions. First, per ethical guidelines (ARRIVE/ARROW/ Association for the Study of Animal Behaviour / Animal Behavior Society Guidelines for the Use of Animals in Research), what was the fate of the fish at the end of the experiment? Second, along with Reviewer 1, I would be interested in seeing more of the Supplementary Methods integrated into the body of the main paper. How were the intruder females housed, maintained, and handled? How many intruder females were used? Were the same ones used repeatedly? If so, was their ID statistically controlled for? In the figure, as the predictions are more related to gender than dominance, I believe the males (females) should be next to each other rather than grouping by dominance. In the figures, the significance values (and associated comparison lines) should not be overlapping with the data – extending up the axis as needed to accommodate the comparisons above the range of the data would help legibility. Jittering the data slightly will prevent individual data points from overlapping and thus also help with legibility (and the jittering should then be mentioned in the figure legends). The figure legends report that the black dots are “residuals”, but I suspect they are the actual data as residuals always average out to zero and these black dots have averages well above zero. The authors may also wish to consider switching colors/symbols to indicate different grouping factors in different plots. The reviewers mentioned further linking the two experiments – would it be possible for the authors to consider looking at individual differences across the two experiments?

Reviewer(s)' Comments to Author:

Referee: 1

Comments to the Author(s)

Goncalves & Radford investigated the within-group consequences of territorial intrusions, and territory defense, against unfamiliar conspecific intruders in the cooperatively breeding cichlid fish *Neolamprologus pulcher*. This was a well-designed study that took an original next step in the study of the relationship between sociality and territory defense. Other studies in this and other systems have investigated the differential response by group members to intruders

depending on relative size, status, and sex, but this study is amongst the first to investigate the impact of this defense and variation in it on within-group interactions. While I had several questions regarding the methods and manuscript (see below), I enjoyed reading this well written study, and look forward to seeing future work building upon these findings.

1. The predictions outlined in lines 103-110 make sense in light of “pay to stay” and “pay to reproduce” hypotheses for cooperative breeding in this system (and are supported by the studies the authors reference in this section). Would these predictions hold true under other hypotheses that have been proposed in this system such as “group augmentation” or “signals of prestige”?
2. While its clearly outlined in the supplemental materials, I’d encourage the authors to explicitly state in the methods section that size-matching of fish was based on SL rather than mass.
3. Even though it is somewhat obvious, it might be worthwhile to describe the “intrusion period” and “post intrusion period” as “... observation period” to make it clear this is when the behavioral observations took place.
4. Did these intrusions take place in the groups home tank, or were fish moved for the trial?
5. The authors classified “bumping” as an affiliative behavior. While to my knowledge this isn’t resolved, some other researchers have considered bumping a submissive or conciliatory behavior. How frequent were bumps relative to other behaviors classified as affiliative? Given that the design of Experiment 2 involves preventing a subordinate female from participating in care, would these other perspectives regarding the context of bumping impact your conclusions?
6. The dominant male and female were very close in size in these groups. In wild groups, males are generally noticeably larger than females. Additionally, Ligocki et al, 2019, (JEZ-A) found that female size relative to their mates influenced female participation in defense. While size differences appear to have remained consistent across groups, could the fact that males were small relative to dominant females have impacted their involvement in defense? Would dominant or subordinate females be more impacted by the size of males in terms of behavioral compensation?
7. It appears that subordinate females varied much more in their participation in territory defense than other individuals. Was there anything that stood out about individuals who were especially territorial against intruders?
8. What were subordinate females doing during the uncooperative treatment? Were they at the barrier “trying” to defend? Was there variation in this behavior that may relate to post-intruder interactions?
9. On line 403 it appears there is an extra space before “Bruintjes...”
10. Were groups kept together following the observation period? Do you have any insights into the long term consequences of these interactions for group members?

Referee: 2

Comments to the Author(s)

The study „Experimental evidence that intruder and group member attributes affect out-group defense and associated within-group interactions in a social fish“ describes two very interesting experiments creating different out-group threats (experiment 1) and different options for

subordinate group members to participate (or not) in territory defense (experiment 2). In general, I think the two experiments are very interesting and enhance our understanding about individual contributions during out-group encounters. Nevertheless I feel that the presentation could be improved particularly to point out the new insights gained from these studies.

Comments

Introduction

In my point of view the introduction does not make it clear how this study differs from previously conducted studies mainly in the same species. For example, in LL47-49 the authors state that “there are relatively few experiments that have tested the causes and magnitude of variation in out-group defensive contributions [...]”. The cited studies are mainly experiments using *N. pulcher* and I think it would make sense to clearly specify what has been done in this species and what aspects are needed to fully understand individual contribution during defensive interactions. This would make it easier to understand the predictions outlined in LL114-126. It’s not clear which predictions are novel and which ones are based on previous experiments. Furthermore some of the predictions (e.g. 2, and 3) are not directly connected to an out-group context because compensation of other group members should only occur if the lost help is indeed beneficial (as also discussed in the disunion) and aggression towards uncooperative subordinates requires that helping is somehow enforced (e.g. pay-to-stay). Both are existent in this system but it’s not entirely clear to me how this is linked with an out-group threat.

After reading the manuscript it is not entirely clear to me why the two experiments have been presented in combination. I think that the authors should make it clearer what are the expectations and predictions of each experiment and how each experiment advances our knowledge of out-group threats.

Results

The results section was extremely difficult to read and I was honestly struggling to link it with the predictions or understand why each test and analysis has been done. I would suggest to structure the results section according to the predictions and analyze each prediction separately instead of combining the results into the experimental time line (pre-intrusion, intrusion, post-intrusion).

In general it is not clear to me why the authors analyzed received and directed behaviors separately for the whole group. In my point of view this would require to correct for multiple testing. Also, I feel that a good justification should be given to analyse the results on a group level and then again more specifically on an individual level (i.e. interactions between DF and SF and vice versa).

LL207-209: Unclear how this was analysed.

LL233-235: Submission is normally a response to received aggression and thus it would be interesting to include received aggression as a covariate in the models.

LL209-2012: In general, did the authors use any correction for multiple testing when performing the posthoc- t tests?

LL310-314: I was surprised that DF affiliation was reduced in the unaware treatment. In Hamilton et al 2005 affiliation e.g. soft touches have been interpreted in an aggressive context. Are affiliative behaviours directed from DF to SF and affiliative behaviours directed from SF to DF the same?

Discussion:

Again I think a clear emphasis on what’s new and what’s different to other experiment is missing throughout the discussion.

LL379-384: In my opinion if DF aggression is used to increase immediate defensive efforts in SF then SF would have also attacked the larger intruder more than any other intruder. However each group member attacked the intruder which posed the highest risk to themselves supporting that higher DF aggression towards SF might be a result of a higher aggressive motivation in DF.

Author's Response to Decision Letter for (RSPB-2019-1261.R0)

See Appendix A.

Decision letter (RSPB-2019-1261.R1)

02-Sep-2019

Dear Dr Braga Goncalves:

Your manuscript has now been peer reviewed and the reviews have been assessed by an Associate Editor. The reviewers' comments (not including confidential comments to the Editor) and the comments from the Associate Editor are included at the end of this email for your reference. As you will see, there are still some concerns with your manuscript. While we are generally pleased with the revisions that have been made, there are still some points that have been detailed by the Associate Editor. We therefore would like to allow you one more round of revision to completely address the issues raised.

This issues could have been dealt with in the latest revision, so we ask that you make every effort to fully address all of the comments at this stage. These revisions will be assessed by the Associate Editor to determine whether you have appropriately addressed them at this final opportunity.

Research ethics:

Use of animals and field studies:

If your study uses animals please include details in the methods section of any approval and licences given to carry out the study and include full details of how animal welfare standards

were ensured. Field studies should be conducted in accordance with local legislation; please include details of the appropriate permission and licences that you obtained to carry out the field work.

If you wish to submit your data to Dryad (<http://datadryad.org/>) and have not already done so you can submit your data via this link [http://datadryad.org/submit?journalID=RSPB&manu=\(Document not available\)](http://datadryad.org/submit?journalID=RSPB&manu=(Document%20not%20available)), which will take you to your unique entry in the Dryad repository.

Please submit a copy of your revised paper within three weeks. If we do not hear from you within this time your manuscript will be rejected. If you are unable to meet this deadline please let us know as soon as possible, as we may be able to grant a short extension.

Best wishes,
Dr Daniel Costa
Editor, Proceedings B
mailto: proceedingsb@royalsociety.org

Associate Editor

Board Member

Comments to Author:

I wish to thank the authors for their careful attention to the reviewer's questions and suggestions. I believe that the paper is much improved, but I still have a few remaining concerns.

1) As both myself and Reviewer 2 had questions about the "intruder" fish, I believe that the authors response to our questions belong in the main document, not the supplementary material: "In Experiment I, we used 30 females for the 36 simulated intrusions (1 dominant-sized, 3 subordinate-sized and 2 small females were used twice each). This information has been added to the Supplementary Methods (lines S48-S50). As only a small proportion of the females were used more than once, we have not statistically controlled for this. In Experiment II, no female was used as an intruder for more than one focal group; the same female was used as the intruder for the different trials to the same focal group (lines S61-S63), but treatment order was counterbalanced and was controlled for statistically (lines S110-S112)."

2) I appreciate the changes the authors made to the figures – please note that the resolution is low and the images appear to be distorted (though of course this might just be a temporary formatting issue, but does need to be resolved at some stage).

3) Reviewer 1, comment #5 – I found this to be a valuable clarification about the bumping behavior and believe that it should be included in the main document. It comes up again in the response to Reviewer 2, comment on LL310-314 and, appears to be somewhat in contrast to the response to Reviewer 1. While I appreciate that the "bumping" behavior is rare in this study, it is attracting some questioning and thus warrants further exposition in the main document.

4) Reviewer 2, comment #1 – While the authors have made good progress in addressing the issues of novelty in the opening paragraph and filled out their study predictions, I think their response to this comment is somewhat lacking in two regards. First, I think the "why this matters" is still not sufficiently clear in the first paragraph – how does assessing outgroup defense and ingroup social interactions *concurrently* change, moderate, or uniquely fill in our understanding of behavioral biology. Second, and relatedly, I think the answer to the "why this matters" question can also be improved by being more explicit about which predictions/questions are novel to this new study setting and which follow from traditional work (as the reviewer initially requested to be distinguished).

5) Reviewer 2, comment on LL379-384. The authors engage in an interesting discussion here that is likely to be of interest to a wider audience.

Author's Response to Decision Letter for (RSPB-2019-1261.R1)

See Appendix B.

Decision letter (RSPB-2019-1261.R2)

17-Sep-2019

Dear Dr Braga Goncalves

I am pleased to inform you that your manuscript entitled "Experimental evidence that intruder and group member attributes affect outgroup defence and associated within-group interactions in a social fish" has been accepted for publication in Proceedings B.

Open Access

Paper charges

Sincerely,

Dr Daniel Costa

Associate Editor:

Board Member

Comments to Author:

I am satisfied with the authors' responses to the final comments, especially where they specified the novel study predictions and broader rationale for the value of this work.

Appendix A

Dear Editor,

Thank you for your positive outlook on our work and for allowing us to resubmit our manuscript 'Experimental evidence that intruder and group member attributes affect outgroup defence and associated within-group interactions in a social fish' (RSPB-2019-1261). We greatly appreciate the helpful suggestions made by the reviewers and the board member. In addressing all the comments (responses provided in bold, with line numbers referring to highlighted sections in the revised manuscript and Supplementary Methods), we believe that we have strengthened the paper further.

We hope that we have addressed all the comments satisfactorily and that you find the manuscript suitable for publication.

Kind regards,
Ines Braga Goncalves and Andy Radford

BOARD MEMBER

First, per ethical guidelines (ARRIVE/ARROW/Association for the Study of Animal Behaviour / Animal Behavior Society Guidelines for the Use of Animals in Research), what was the fate of the fish at the end of the experiment?

All fish were retained as part of the captive study population at the end of each experiment (lines 192–193).

Second, along with Reviewer 1, I would be interested in seeing more of the Supplementary Methods integrated into the body of the main paper. How were the intruder females housed, maintained, and handled? How many intruder females were used? Were the same ones used repeatedly? If so, was their ID statistically controlled for?

For each experiment, intruder females were selected from other (focal and non-focal) social groups; females were never used as an intruder and as part of a focal social group in the same week (main document lines 189–191; Supplementary Methods lines S50–S52). Prior to each trial, an intruding female was netted out of her tank and held in a small container with home-tank water for 10 min before being introduced into the intruder compartment in the focal tank and left to settle for another 5 min. After the intrusion, the female was netted out of the compartment and transferred back to her home tank. This information is provided in lines 162–168.

In Experiment I, we used 30 females for the 36 simulated intrusions (1 dominant-sized, 3 subordinate-sized and 2 small females were used twice each). This information has been added to the Supplementary Methods (lines S48–S50). As only a small proportion of the females were used more than once, we have not statistically controlled for this. In Experiment II, no female was used as an intruder for more than one focal group; the same female was used as the intruder for the different trials to the same focal group (lines S61–S63), but treatment order was counterbalanced and was controlled for statistically (lines S110–S112).

In the figures, as the predictions are more related to gender than dominance, I believe the males (females) should be next to each other rather than grouping by dominance. In the figures, the significance values (and associated comparison lines) should not be overlapping with the data—extending up the axis as needed to accommodate the comparisons above the range of the data would help legibility. Jittering the data slightly will prevent individual data points from overlapping and thus also help with legibility (and the jittering should then be mentioned in the figure legends). The figure legends report that the black dots are “residuals”, but I suspect they are the actual data as residuals always average out to zero and these black dots have averages well above zero. The authors may also wish to consider switching colors/symbols to indicate different grouping factors in different plots.

In figures 1 to 3, we have rearranged the order of the individual categories so that they are grouped by sex rather than dominance, extended the y-axes and shifted the significance symbols to aid legibility, and changed the colours as suggested. The datapoints (black dots) in the plots are partial residuals around the fitted mean for each treatment/individual category combination; we have clarified in the figure legends that the points represent partial residuals (lines 658, 666 and 673). We could not find a way of successfully jittering the datapoints for each category of the second factor (described in the figure legend). Our attempts led to jittering of all points within each category of the main factor on the x-axis, which was unhelpful and created lack of clarity. However, since each “column” of data has only 12 or 13 values, data points are not ‘lost’ from view and we believe the figures are clear in this respect; all data will also be made available via Dryad.

The reviewers mentioned further linking the two experiments—would it be possible for the authors to consider looking at individual differences across the two experiments? **We do not have the data to analyse individual differences across the two experiments. To increase the number of social groups where each dominant was at least 5 mm larger than the same-sex subordinate, we separated some of the groups used in Experiment I and used those individuals with new ones to form new groups for Experiment II; we did not keep track of individual fish between the two studies. This is a subject certainly worth considering in the future.**

REFEREE 1

1. The predictions outlined in lines 103-110 make sense in light of “pay to stay” and “pay to reproduce” hypotheses for cooperative breeding in this system (and are supported by the studies the authors reference in this section). Would these predictions hold true under other hypotheses that have been proposed in this system such as “group augmentation” or “signals of prestige”?

This is an interesting question. Neither the receipt of rewards in exchange for helping effort nor punishment are predicted by the “signals of prestige” or by the “group augmentation” hypotheses. A net benefit of helping to dominants is not predicted by signals of prestige but is predicted by group augmentation; therefore, we might expect social monitoring under the latter but not the former hypothesis. Overall, our expectations of the social dynamics between dominant females (DFs) and subordinate females (SFs) are best supported by the “pay-to-stay” or “pay-to-reproduce” hypotheses, which is why these are explicitly mentioned in the Introduction.

2. While its clearly outlined in the supplemental materials, I'd encourage the authors to explicitly state in the methods section that size-matching of fish was based on SL rather than mass.

We have added this information to the Methods (lines 148–149).

3. Even though it is somewhat obvious, it might be worthwhile to describe the “intrusion period” and “post intrusion period” as “... observation period” to make it clear this is when the behavioral observations took place.

We have made this change as suggested (e.g. lines 162–168, 195 and 205).

4. Did these intrusions take place in the groups home tank, or were fish moved for the trial?

All intrusions took place in the groups' home tanks. We created an intruder compartment by sliding down a transparent partition 8 cm from one edge of the focal tank, so that the intruder was in the focal group's territory and individuals could interact with each other visually but not physically (main document lines 162–165; Supplementary Methods lines S25–S28).

5. The authors classified “bumping” as an affiliative behavior. While to my knowledge this isn't resolved, some other researchers have considered bumping a submissive or conciliatory behavior. How frequent were bumps relative to other behaviors classified as affiliative? Given that the design of Experiment 2 involves preventing a subordinate female from participating in care, would these other perspectives regarding the context of bumping impact your conclusions?

We are not aware of any formal classification in the literature of “bumping” as a submissive behaviour in this species. Two of the most oft-cited ethograms for *N. pulcher* describe “bumps” (also known as “soft touches”) either as a *non-aggressive and social behaviour* [1] or as an *affiliative behaviour* [2]. Other oft-cited ethograms of *N. pulcher* that focus on submissive behaviours [3–5] do not include “bumps” in their repertoire. Furthermore, a number of empirical studies that base their behavioural categorisations on the same ethograms we have used, also describe “bumps/soft touches” as an affiliative behaviour [6–9]. Hamilton et al. (2005) refer to personal communications when speculating whether subordinate bumps towards dominant individuals should be considered affiliative or submissive, but their principal component analysis places “subordinate bumps displayed” clearly in the first principal component which was interpreted as an affiliative axis, and these authors have interpreted “bumps” as affiliative displays in more recent publications [e.g. 9]. We therefore think it most appropriate to leave the classification of “bumps” as an affiliative behaviour.

In any case, “bumps” were relatively rare. In Experiment II's Uncooperative treatment, where we inhibited SF ability to take part in territorial defence, we observed very low levels of post-intrusion subordinate bumping towards the DF (3.1% of the total affiliative behaviour). This result is against expectations if SF bumping has a conciliatory or submissive function. Bumps also made up a very small proportion of DF affiliation towards SFs (0.5%). Thus, the inclusion of bumps in our affiliative behaviour category is highly unlikely to drive our findings and conclusions and so we have retained the classification as per the original submission of the manuscript (Supplementary Methods lines S84–S85).

6. The dominant male and female were very close in size in these groups. In wild groups, males are generally noticeably larger than females. Additionally, Ligocki et al, 2019, (JEZ-A) found that female size relative to their mates influenced female participation in defense. While size differences appear to have remained consistent across groups, could the fact that males were small relative to dominant females have impacted their involvement in defense? Would dominant or subordinate females be more impacted by the size of males in terms of behavioral compensation?

Our dominant pairs were indeed closer in body length than usually reported in other laboratory studies. However, considering the behaviours that have been examined before, the patterns we found in our study (greater defensive effort by DFs and SFs towards same-sex size-matched intruders, little defensive effort by dominant males (DMs) towards opposite-sex intruders) are in-line with those reported previously [3,11]; we make this clear in our Discussion (lines 382–385). Furthermore, as in Ligocki et al. (2019), we find similar levels of defence by DMs and DFs against smaller intruders (this manuscript, Figure 1) despite their small size difference. We have no reason to believe that relative DM size affects SF defence behaviour, particularly as our intruders were all females and thus DMs were expected to (and did; Figure 1) participate less strongly in defence.

7. It appears that subordinate females varied much more in their participation in territory defense than other individuals. Was there anything that stood out about individuals who were especially territorial against intruders?

We did not notice any abnormal behaviour by any of our experimental subjects during the experiment; there were no clear-cut stand-out behaviours or reasons for this variation. Moreover, SF defensive behaviour was not strongly correlated with either their own body size ($r_s = 0.239$, $p = 0.160$) or with the size difference between the DF and SF in each group ($r_s = 0.269$, $p = 0.113$).

8. What were subordinate females doing during the uncooperative treatment? Were they at the barrier “trying” to defend? Was there variation in this behavior that may relate to post-intruder interactions?

We have not quantified SF behaviour in the Uncooperative treatment because our focus was how individual defensive contributions changed in response to the treatments (lines 158–160), and SFs could not take part in defence in this treatment (lines 180–182). However, all SFs spent time watching the intrusions and their group members, and all attempted to join their group by swimming alongside the barrier and nudging against it (lines 182–184). It is not possible for us to judge whether these females were trying to defend against the intruder.

9. On line 403 it appears there is an extra space before “Bruintjes...”

We have removed the extra space (line 436).

10. Were groups kept together following the observation period? Do you have any insights into the long term consequences of these interactions for group members?

Although some groups were kept together at the end of the experimental trials, we did not carry out further behavioural observations. We are currently analysing the results of an experiment that investigated the long-term behavioural and reproductive consequences of repeated territorial intrusions during a 3-month period; this work will be prepared for a separate paper.

REFEREE 2

In my point of view the introduction does not make it clear how this study differs from previously conducted studies mainly in the same species. For example, in LL47-49 the authors state that “there are relatively few experiments that have tested the causes and magnitude of variation in out-group defensive contributions [...]”. The cited studies are mainly experiments using *N. pulcher* and I think it would make sense to clearly specify what has been done in this species and what aspects are needed to fully understand individual contribution during defensive interactions. This would make it easier to understand the predictions outlined in LL114-126. It’s not clear which predictions are novel and which ones are based on previous experiments.

We have structured our Introduction such that the first paragraph gives an overview of the research field and what is novel about our work, and subsequent paragraphs provide detailed background on what is already known from other species, including *N. pulcher*. So, the detail about previous experiments and studies on *N. pulcher* are provided in those subsequent paragraphs, even though there are some initial citations of relevant references in the first paragraph.

Our study has a number of novel aspects, which we have made clearer in the first Introduction paragraph (lines 48–51). With respect to defence against intruders, relatively few experimental studies have assessed the causes and magnitude of individual variation in contributions, and most of these studies assessed the behaviour of a subset of, rather than all, group members. With respect to how outgroup threats affect within-group behaviour, there have been no experimental studies considering those social interactions during contests with outsiders; the only previous research was observational work in vervet monkeys. There have been a small handful of experimental studies considering some post-contest changes in within-group behaviour, including one on *N. pulcher*, but these have generally considered overall changes in behaviour, rather than comparing variation arising due to differences in intruder identity or group member contributions to defence. There have been no experimental studies in any species that have considered all of these different stages and aspects together.

To aid clarity about what has been previously done in this research field, we provide species names for studies that we cite in paragraphs 2–4 of the Introduction (e.g. lines 58, 64, 70, 76 and 81). Moreover, we include two entire paragraphs on *N. pulcher* (lines 88–117), providing both background information of relevance to later predictions and detailing previous work of direct relevance to the current experiments; the latter are explicitly stated in lines 93–97 and 104–106. Because communal territorial defence is a key cooperative behaviour in this species, several studies have assessed how group member attributes (e.g. status, sex and size) and intruder attributes (e.g. size and sex) influence defensive efforts in at least some group members. Similarly, a few studies have investigated how social groups respond to the temporary absence and return of a group member, including in an outgroup context situation. We build on this work by additionally exploring how all group members respond to different intruders, and how they interact with each other both during and in the aftermath of outgroup contests. In the final paragraph of the Introduction (lines 123–143), we cite references (including those on *N. pulcher*) on which our predictions are based.

Furthermore some of the predictions (e.g. 2, and 3) are not directly connected to an out-group context because compensation of other group members should only occur if the lost help is indeed beneficial (as also discussed in the disunion) and aggression towards uncooperative subordinates requires that helping is somehow enforced (e.g. pay-to-stay). Both are existent in this system but it's not entirely clear to me how this is linked with an out-group threat.

We are not entirely sure what the issue is here. As the reviewer states, the relevant behaviours exist in this system, and we are examining defensive actions and related within-group interactions in response to territorial intrusions by a conspecific (a clear outgroup threat). By definition, therefore, there is a connection to an outgroup context.

After reading the manuscript it is not entirely clear to me why the two experiments have been presented in combination. I think that the authors should make it clearer what are the expectations and predictions of each experiment and how each experiment advances our knowledge of out-group threats.

We believe that the two studies complement each other by examining two key factors expected to influence participation in both outgroup contests and associated within-group interactions (lines 118–122). We have clarified which predictions refer to which experiment in both the Introduction (lines 123–143) and in the restructured Results (lines 237, 250, 256, 288 and 303). Our restructuring of the Results (see below), means that findings from the two experiments are combined more obviously, as they are also in the Discussion.

The results section was extremely difficult to read and I was honestly struggling to link it with the predictions or understand why each test and analysis has been done. I would suggest to structure the results section according to the predictions and analyze each prediction separately instead of combining the results into the experimental time line (pre-intrusion, intrusion, post-intrusion).

We agree that there could have been greater clarity with respect to which results link with which predictions. However, restructuring the Results section by prediction alone creates its own issues. Most notably, we think it is important to present full results in terms of defensive behaviours and within-group interactions from each available time period (that is one of the strengths and novelties of our work; lines 48–51), but it is not possible or practical to make specific predictions about each and every possible combination. For instance, the paucity of previous work on within-group interactions during outgroup contests means there is little previous literature to draw on to make specific predictions. Instead, we explain in the final paragraph of the Introduction that we are considering all group members and behaviours, but are also making, where possible, some specific predictions grounded in previous work (lines 122–123).

To aid clarity, we have therefore done the following. We have ensured that all sections of the MS follow the same general order (i.e. considering defensive behaviour and then associated within-group interactions, both during and post-contest). That is, for example, how we had structured the opening paragraph of the Introduction, the ordering of subsequent Introduction paragraphs and the Discussion in the original submission (and continue to do so in the current version). But, now, we also have the predictions in the final paragraph of the

Introduction ordered this way and have reordered the Results section accordingly: rather than giving all the results of each experiment in turn, split by time period, we have split the results by time period with each experiment included in each case. This also means that the subheadings of the Results and the Discussion are identical. Moreover, we have explicitly numbered the predictions in the final paragraph of the Introduction (lines 123–143), indicated which predictions can be considered from which subsections of the Results (e.g. lines 237, 250 and 256), and referred to specific predictions by number where relevant in the Discussion (lines 385, 406, 434, 461, 463, 470 and 494).

In general it is not clear to me why the authors analyzed received and directed behaviors separately for the whole group. In my point of view this would require to correct for multiple testing. Also, I feel that a good justification should be given to analyse the results on a group level and then again more specifically on an individual level (i.e. interactions between DF and SF and vice versa).

We have not analysed combined behaviour of the group as a whole. Rather, we have analysed contributions of different group members to a given behaviour in each treatment, and the interaction between them (as we expect that there might be different responses by different categories of individual to different treatments in at least some cases) (lines 206–210). The datasets for aggression displayed and aggression received, for example, are therefore different (they are not mirror-images of one another) negating a clear-cut need for corrections of multiple testing. These overall analyses are important because only rarely (if ever) have previous studies considered the behaviour of all group members in these respects. In addition, we are also particularly interested in (justification provided on lines 111–117), and have specific predictions related to (e.g. lines 138–143), the interactions between DFs and SFs, and thus have analysed those data too.

LL207-209: Unclear how this was analysed.

We have added this information to the main text (lines 210–211): when the interactions between treatment and individual category were found to be significant in the main models, we assessed the effects of treatment for each category separately.

LL233-235: Submission is normally a response to received aggression and thus it would be interesting to include received aggression as a covariate in the models.

We agree. However, adding received aggression to our models of displayed submission during intrusions in Experiment I did not qualitatively change our results; neither treatment nor its interaction with individual category had significant effects on submissive displays (see table below). Therefore, for consistency, we have chosen not to add received aggression as a factor in this model in the manuscript.

FINAL MODEL	χ^2	t-value	d.f.	P	estimate	s.e.
Intercept		2.63	63.33	0.011	1.72	0.65
Individual category	16.52		3	<0.001		
DM		2.70	32.92	0.011	2.25	0.83

SF		1.83	39.17	0.075	1.61	0.88
SM		4.15	33.30	<0.001	3.47	0.84
Order	10.87		2	0.004		
Day 2		-2.46	92.92	0.016	-1.17	0.47
Day 3		-3.15	93.76	<0.001	-1.50	0.48
Aggression received		4.95	135.06	<0.001	0.15	0.03
REMOVED TERMS		χ^2	d.f.	P		
Intruder responsiveness x Treatment		2.42	2	0.298		
Aggression received x Treatment		3.90	2	0.142		
Treatment x Individual category		6.47	6	0.373		
Intruder responsiveness		0.27	1	0.604		
Treatment		0.55	2	0.761		

LL209-2012: In general, did the authors use any correction for multiple testing when performing the posthoc- t tests?

No, we did not use correction factors in our post-hoc tests because we selectively conducted post-hoc tests to answer only specific hypotheses related to treatment and treatment-by-individual category effects, not to uncover all possible significant effects of all factors.

LL310-314: I was surprised that DF affiliation was reduced in the unaware treatment. In Hamilton et al 2005 affiliation e.g. soft touches have been interpreted in an aggressive context. Are affiliative behaviours directed from DF to SF and affiliative behaviours directed from SF to DF the same?

Hamilton et al. (2005) predicted that increased within-group conflict leads to reductions in affiliation, increases in aggression and in submission as well as in avoidance behaviour, and that all measures of conflict are negatively correlated with subordinate helping effort. Our results show that in the Unaware treatment, where SFs are not in sight of the dominants and do not contribute to defence, DFs reduced their affiliation towards SFs, which is line with the predictions from Hamilton et al. (2005). Regarding soft touches/bumping specifically, Hamilton et al. (2005) suggest that “dominant bumps towards subordinates” may occur in aggressive contexts and interpret “bumps received” by subordinates as part of a submissive behaviour axis in a principal component analysis. However, in our results, bumps made up only 0.5% of total DF affiliative acts towards SFs and thus we find it unlikely that bumps play an important aggressive/punitive role in the aftermath of our Unaware treatment.

Again I think a clear emphasis on what’s new and what’s different to other experiment is missing throughout the discussion.

In the first paragraph of the Discussion, we provide an overview of our main findings and then in the final sentence give the overall novelty of our work (lines 372–376), which matches with our description of the main selling points of the paper in the opening paragraph of the Introduction (see earlier response). We also emphasise this novelty in the Concluding paragraph (lines 511–512). In the middle sections of the Discussion, we make it explicitly clear how individual findings of ours relate to previous experimental studies on *N. pulcher* in this

research field (e.g. lines 387–389, 417–421, 434–436). However, it is the overall package of work – considering how all group members contribute to defence and interact socially both during and after intrusions, and considering experimentally how those are affected both by intruder identity and variation in defensive effort – that is the most novel aspect of our paper.

LL379-384: In my opinion if DF aggression is used to increase immediate defensive efforts in SF then SF would have also attacked the larger intruder more than any other intruder. However each group member attacked the intruder which posed the highest risk to themselves supporting that higher DF aggression towards SF might be a result of a higher aggressive motivation in DF.

We respectfully disagree with the reviewer here. First, although group members defended most strongly against intruders that posed a direct threat to their specific social position in the group, all group members showed some defensive behaviour against all intruders. Second, if DF aggression functions to increase SF defensive efforts, then SFs should increase their defensive efforts relative to when such aggression is absent, and not in absolute value relative to all treatments. DFs may have been more aggressive towards SFs to extract greater defensive action from SFs and because when facing a size-matched intruder they become more aggressively motivated; the two hypotheses are not mutually exclusive.

References:

- [1] Sopinka, N.M., Fitzpatrick, J.L., Desjardins, J.K., Stiver, K.A., Marsh-Rollo, S.E. & Balshine, S. 2009 Liver size reveals social status in the African cichlid *Neolamprologus pulcher*. *Journal of Fish Biology* **75**, 1-16. (doi:10.1111/j.1095-8649.2009.02234.x).
- [2] Reddon, A.R., O'Connor, C.M., Marsh-Rollo, S.E., Balshine, S., Gozdowska, M. & Kulczykowska, E. 2015 Brain nonapeptide levels are related to social status and affiliative behaviour in a cooperatively breeding cichlid fish. *Royal Society Open Science* **2**, 140072. (doi:10.1098/rsos.140072).
- [3] Ligocki, I.Y., Reddon, A.R., Hellmann, J.K., O'Connor, C.M., Marsh-Rollo, S., Balshine, S. & Hamilton, I.M. 2015 Social status influences responses to unfamiliar conspecifics in a cooperatively breeding fish. *Behaviour* **152**, 1822-1840. (doi:10.1163/1568539x-00003306).
- [4] Buchner, A.S., Sloman, K.A. & Balshine, S. 2004 The physiological effects of social status in the cooperatively breeding cichlid *Neolamprologus pulcher*. *Journal of Fish Biology* **65**, 1080-1095. (doi:10.1111/j.1095-8649.2004.00517.x).
- [5] Hick, K., Reddon, A.R., O'Connor, C.M. & Balshine, S. 2014 Strategic and tactical fighting decisions in cichlid fishes with divergent social systems. *Behaviour* **151**, 47-71. (doi:10.1163/1568539x-00003122).
- [6] Mileva, V.R., Fitzpatrick, J.L., Marsh-Rollo, S., Gilmour, K.M., Wood, C.M. & Balshine, S. 2009 The stress response of the highly social African cichlid *Neolamprologus pulcher*. *Physiol. Biochem. Zool.* **82**, 720-729. (doi:10.1086/605937).
- [7] Reddon, A.R., O'Connor, C.M., Marsh-Rollo, S.E. & Balshine, S. 2012 Effects of isotocin on social responses in a cooperatively breeding fish. *Anim. Behav.* **84**, 753-760. (doi:10.1016/j.anbehav.2012.07.021).
- [8] Bruintjes, R., Lynton-Jenkins, J., Jones, J.W. & Radford, A.N. 2016 Out-group threat promotes within-group affiliation in a cooperative fish. *American Naturalist* **187**, 274-282. (doi:10.1086/684411).

- [9] Hellmann, J.K. & Hamilton, I.M. 2019 Intragroup social dynamics vary with the presence of neighbors in a cooperatively breeding fish. *Current Zoology* **65**, 21-31. (doi:10.1093/cz/zoy025).
- [10] Hamilton, I.M., Heg, D. & Bender, N. 2005 Size differences within a dominance hierarchy influence conflict and help in a cooperatively breeding cichlid. *Behaviour* **142**, 1591-1613. (doi:10.1163/156853905774831846).
- [11] Desjardins, J.K., Stiver, K.A., Fitzpatrick, J.L. & Balshine, S. 2008 Differential responses to territory intrusions in cooperatively breeding fish. *Anim. Behav.* **75**, 595-604. (doi:10.1016/j.anbehav.2007.05.025).
- [12] Ligocki, I.Y., Earley, R.L. & Hamilton, I.M. 2019 How individual and relative size affect participation in territorial defense and cortisol levels in a social fish. *J. Exp. Zool.* **331**, 217-226. (doi:doi.org/10.1002/jez.2255).

Appendix B

Dear Editor,

Thank you for the overall positive assessment of our revised manuscript ‘Experimental evidence that intruder and group member attributes affect outgroup defence and associated within-group interactions in a social fish’ (RSPB-2019-1261.R1), and the recognition that we had paid careful attention to the original reviews and suggestions. The main reason that even more additional information had not previously been added to the manuscript was that we had reached the journal’s page limit. However, where necessary, we have found ways to include the information requested in this latest set of comments (responses provided in bold, with line numbers referring to highlighted sections in the revised manuscript and Supplementary Methods).

We hope that we have addressed all the comments satisfactorily and that you find the manuscript suitable for publication.

Kind regards,
Ines Braga Goncalves and Andy Radford

BOARD MEMBER

1) As both myself and Reviewer 2 had questions about the “intruder” fish, I believe that the authors response to our questions belong in the main document, not the supplementary material: “In Experiment I, we used 30 females for the 36 simulated intrusions (1 dominant-sized, 3 subordinate-sized and 2 small females were used twice each). This information has been added to the Supplementary Methods (lines S48–S50). As only a small proportion of the females were used more than once, we have not statistically controlled for this. In Experiment II, no female was used as an intruder for more than one focal group; the same female was used as the intruder for the different trials to the same focal group (lines S61–S63), but treatment order was counterbalanced and was controlled for statistically (lines S110–S112).”

As requested, we have added the relevant information to the main document. For Experiment I, we have included “We used 30 females for the 36 simulated intrusions (one Large, three Medium and two Small females were used as intruders twice each).” in the main document Methods when describing the treatments for the first time (lines 181–182). For Experiment II, we have included “no female was used as an intruder for more than one focal group, but each group received the same female intruder in all three trials.” in the main document Methods when describing the treatments for the first time (lines 185–187). We state in the main document Methods that treatment order was counterbalanced in both experiments (lines 170–171). Since information on our statistical models, including the specific factors included in each case, are in the Supplementary Methods, our view is that this is the most relevant place for inclusion of “We did not control for intruder identity because only a small proportion of female intruders in Experiment I were used more than once, and no female in Experiment II was used as an intruder for more than one focal group.” (lines S119–S121).

2) I appreciate the changes the authors made to the figures—please note that the resolution is low and the images appear to be distorted (though of course this might just be a temporary formatting issue, but does need to be resolved at some stage).

Thank you. We have addressed these remaining figure-quality issues.

3) Reviewer 1, comment #5 – I found this to be a valuable clarification about the bumping behavior and believe that it should be included in the main document. It comes up again in the response to Reviewer 2, comment on LL310-314 and, appears to be somewhat in contrast to the response to Reviewer 1. While I appreciate that the “bumping” behavior is rare in this study, it is attracting some questioning and thus warrants further exposition in the main document.

As requested, we have added extra information regarding bumping to the paper, thus providing the valuable clarification about its categorisation in the way that we had only done previously in the response document. However, because we are interested in and analyse the broad behavioural categories of aggression, submission and affiliation (lines 207–211), rather than the individual behaviours included within each category (which we describe in the Supplementary Methods), we have added the extra information on bumping to the Supplementary Methods (lines S84–S88). Bumping behaviour attracted some original questioning by Referee 1 because they claimed that “some other researchers have considered bumping a submissive or conciliatory behaviour”. However, the referee provided no citations to support this claim and, as we detailed in our original response, all the papers that we can find that formally classify bumps in the study species have it as an affiliative behaviour; as we now detail in the paper (lines S84–S88). The only possible reference to bumps as submissive behaviour that we can find is a personal communication speculating about this in Hamilton et al. (2005), but in that paper the authors’ own principal component analysis clearly places subordinate bumps displayed in the affiliation category, and those authors have subsequently interpreted bumps as affiliative in more recent papers (e.g. Hellmann & Hamilton 2019). Similarly, Referee 2’s suggestion that dominant bumps received by subordinates may take place in aggressive contexts is based just on the personal observations reported in Hamilton et al. (2005). Moreover, this suggestion contrasts our own results of reduced (rather than increased) DF affiliation (in which bumps are included) towards SFs following defective behaviour. Given the lack of any clear evidence to classify bumps as anything other than affiliative in this species, we think that it would be inappropriate to have a discourse on this one behavioural element in our main paper document.

4) Reviewer 2, comment #1 – While the authors have made good progress in addressing the issues of novelty in the opening paragraph and filled out their study predictions, I think their response to this comment is somewhat lacking in two regards. First, I think the “why this matters” is still not sufficiently clear in the first paragraph—how does assessing outgroup defense and ingroup social interactions *concurrently* change, moderate, or uniquely fill in our understanding of behavioral biology. Second, and relatedly, I think the answer to the “why this matters” question can also be improved by being more explicit about which predictions/questions are novel to this new study setting and which follow from traditional work (as the reviewer initially requested to be distinguished).

As requested, we have included in the first paragraph of the Introduction further clarification of why it is important to study how outgroup conflict can influence ingroup social interactions in general and why it is important to study defensive actions and ingroup interactions concurrently (lines 41–45). This is in addition to the existing material explaining the importance of our experimental study (e.g. lines 53–55, 383–386). Also as requested, we have included explicit statements about novelty relating to our predictions in the final paragraph of the Introduction (lines 129–132, 133–135, 139–140, 141–143). This is in addition to clear indications of how our work relates directly to previous studies given throughout the Discussion (e.g. lines 397–399, 429–431, 447–449).

5) Reviewer 2, comment on LL379-384. The authors engage in an interesting discussion here that is likely to be of interest to a wider audience.

We agree and have edited this point in the Discussion in two ways. First, we acknowledge that further experiments would be needed to test explicitly whether dominant aggression towards subordinates during intrusions does function as a social incentive to increase subordinate participation in territorial defence; i.e. whether relative changes in defensive contributions to the same intruder are influenced by within-group behavioural interactions (lines 426–427). Second, we have included the alternative possibility suggested by Reviewer 2 that higher dominant aggression towards the subordinate during intrusions may also result from higher aggressive motivation due to the presence of large intruders (lines 427–429).